# Cross-Region Comparison Intellectual Capital and Its Impact on Islamic Banks Performance

**Prasojo Prasojo** [1,2,*] ⬤, **Winwin Yadiati** [1], **Tettet Fitrijanti** [1] **and Memed Sueb** [1]

[1] Faculty of Economics and Business, Padjadjaran University, Bandung 40132, Indonesia;
winwin.yadiati@unpad.ac.id (W.Y.); tettet.fitrijanti@fe.unpad.ac.id (T.F.);
memed.sueb@fe.unpad.ac.id (M.S.)

[2] Faculty of Economics and Business Islam, State Islamic University Sunan Kalijaga,
Yogyakarta 55281, Indonesia

[*] Correspondence: prasojo@uin-suka.ac.id

**Abstract:** This paper uses the value-added intellectual coefficient (VAIC) to assess the performance of Islamic banks (IBs) by measuring return on assets (ROA) and income from financing Islamic banks (IFIB). The model tests the relationship between intellectual capital (IC) and IB performance in various regions using a panel data regression methodology with a fixed-effects model and IB financial data for the period 2009–2019 from the BankScope database. The empirical results show that VAIC has a significant positive effect on IB performance using both ROA and IFIB proxies. Furthermore, human capital and capital employed efficiency have a positive relationship with ROA and IFIB, while structural capital efficiency has a relationship with ROA, but is not related to IFIB. The results can be used by companies in strategic decision making related to IC, especially human capital, structural capital, and employed capital.

**Keywords:** Islamic bank; ROA; IFIB; intellectual capital; VAIC

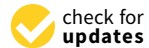



## 1. Introduction

Resource theory claims that the composition of tangible and intangible resources managed by a company is the main factor in its performance. According to Scafarto et al. (2016), intellectual capital (IC) and human capital (HC) are essential to running a business. To maintain sustainable performance, a company must have strategic resources in the form of IC (Hsu and Wang 2012). The development of information technology and science has triggered interest in IC research (Guthrie et al. 2001). IC is an instrument that can affect company performance (Edvinsson and Malone 1997). Therefore, companies need to create new knowledge to secure their future and must have a good strategy and good resource management (Nonaka et al. 2017).

IC must be managed effectively since it plays a vital role in a company's performance and competitiveness. Chang and Hsieh (2011) note that IC is challenging to identify and measure. A company must choose the appropriate method for measuring and determining IC utilisation. Pulic (2000) develops a value-added intellectual coefficient method known as the VAIC. This method is easy to adopt because it is logical, easy to use, and has standards (Bayraktaroglu et al. 2019). The data used to calculate efficiency are from financial reports for comparison between companies (Maditinos et al. 2011). The VAIC method focuses on the transformation of the use of physical capital and IC in generating added value (Pulic 2000).

Islamic banks (IBs) are rooted in Islamic ideology, so all their activities must comply with Islamic law. The special feature of Islamic banks is that there is a prohibition on transactions using interest instruments, and their operational activities must comply with Islamic principles (Neifar and Jarboui 2018). This derived from Islamic law, which prohibits interest-based transactions (riba), speculative activities (gharar), and funding projects that

violate sharia. In addition, IBs must maintain their credibility, reputation, and legitimacy as financial institutions that provide banking services. As financial institutions, IBs serve not only the needs of Muslims, but those from all segments of society; they must provide solutions by utilising IC to create a competitive advantage and improve their performance. Therefore, IBs require a higher level of IC capability, and the combination of HC, SC, and CE encourages innovation that will contribute to value creation (Nawaz and Haniffa 2017). This study is vital in analysing more deeply the following three components, namely, HC, structural capital (SC), and capital employed (CE), which contribute the most to improving the performance of IBs.

IBs are Islamic financial institutions that not only focus on serving the needs of people who meet sharia principles, but which must also benefit the broader society. The measurement of the performance of IBs should ideally include financial dimensions and aspects of sharia (Mutia et al. 2019). Many previous studies link IC to the performance of IBs. However, in general, the performance assessment of IBs only considers the financial dimension. Meanwhile, this research measures the performance of IBs from the financial and Islamic aspects. The financial assessment involves considering these banks' returns on assets (ROA) and returns on equity (ROE). From the perspective of sharia, the assessment considers revenue from the profit-sharing scheme.

Some of the results of previous studies partially relate IC to company performance, but it is not clear if these can be extended beyond the results of those particular studies (Kweh et al. 2019). There are varying results from empirical studies that consider IC components as independent, stand-alone variables (e.g., Nawaz and Haniffa 2017; Tiwari 2021; Vo and Tran 2021). Nawaz and Haniffa (2017) find that SC is not significant to performance. Tiwari and Vidyarthi (2018) find evidence that CE is not significant for performance, while Vo and Tran (2021) find that only CE is proven to affect performance.

The present study investigates the relationships between IC and the performance of IBs from a financial and sharia perspective. It is expected to fill the gap in the literature on IC and performance in IBs in a global context by empirically testing the added value of IC on the performance of IBs around the world using the VAIC method. We examine the IC components partially against their independent variables for a more in-depth analysis. Islamic banking is one of the knowledge-based industries that requires more intellectual property. Therefore, IBs will achieve better performance if they continue to manage ICs more efficiently. IB performance has been explored by many previous researchers using market-based and accounting measurements. This paper attempts to fill the gap by exploring the IC efficiency of IBs and their relationship to performance on the basis of IFIB. The findings of this study will be useful for Islamic banks who want to measure IC efficiency. Thus, it will motivate managers to measure and manage their IC.

This study contributes to the IC literature by examining the impact of IC on the performance of IBs in several ways. First, IC is proxied by the value-added intellectual coefficient (VAIC) and its subcomponents: HC efficiency (HCE), structural capital efficiency (SCE), and CE efficiency (CEE). Second, this study was conducted across countries to increase the generalizability of the findings. Third, the mainstream literature measures performance using accounting and market-based performance, whereas we use the accounting measures ROA and income from financing Islamic banks (IFIB). Finally, this is one of the first studies to explore the empirical relationship of IC with Islamic banks from many countries so that the findings can be more generalised.

## 2. Literature Review

### 2.1. Intellectual Capital and Measurement

Intangible resources, including intellectual property rights, can create a competitive advantage (Hall 1992); therefore, companies must identify intangible resources in order to provide greater performance (Hall 1993). According to Brooking (1996), IC is part of the company's intangible assets. Furthermore, Brooking (1997) emphasises that intangible

assets must be identifiable, documentable, and measurable in order to be easily audited so as to provide optimal utility.

Edvinsson and Sullivan (1996) define IC as the knowledge that can be converted into value. According to Stewart (1998), IC is the accumulation of individual knowledge and the skills, expertise, and knowledge contained in the human brain. Bontis et al. (2000) divide IC into three categories, namely, HC, SC, and CE. HC includes the experience, knowledge, and abilities of employees. SC consists of internal organisational processes and routines that support the business. CEE represents the investments of financial and physical capital, which are important as intangible assets to increase the productivity and performance of Islamic banks (Smriti and Das 2018).

HC is defined by Hsu and Fang (2009) as the total of all individual knowledge, skills, abilities, expertise, and experiences in an entity, which can be utilised to achieve its goals. Lowendahl (1997) divides the category of competence into two sub-groups consisting of individuals and collectives. Litschaka et al. (2006) believe that HC should be assessed in monetary units to increase organisational awareness to maintain HC because this will create success in the future. The efficient use of HC can maximise firm value, and firms must thus increase investment in intangible assets, especially HC (Kharal et al. 2014). This investment will generate knowledge, skills, or other values that cannot be separated from the CE involved in these activities. Companies have the risk of losing HC through the headhunting of employees by competitors (Olander et al. 2015). Islamic bank HCs are required to have more expertise than conventional banks considering that their products are more complex. Therefore, banks need to facilitate skill improvement and special education related to Islamic character considering that the core business of IBs is closely related to Islamic values. HC development is an investment that has a positive impact on the performance of IBs.

SC is an intellectual asset that remains with the company when an employee has left. Generally, it is explicit and does not depend on individuals (Longo et al. 2009). Massaro et al. (2019) define SC as an organisation's explicit knowledge, including all non-human knowledge resources in the organisation, such as databases, organisational charts, process manuals, strategies, routines (Bontis et al. 2000), patents, licences, and trademarks, that are of material value to the company (Chen et al. 2006; Hirsch-Kreinsen et al. 2006; Pike et al. 2005; Roos and Roos 1997). Organisations with substantial SC will have a culture that supports individuals trying new things and always learning from failure; this provides an essential link to IC that allows it to be measured at the organisational level (Bontis et al. 2000). CE shows the added value of each nominal investment towards capital creation, as well as the efficiency of the employment of physical and financial capital (Kweh et al. 2019). The critical value of CE is shown in the relationship between the importance of investment in IC compared to financial planning and budgeting. In the context of IB, CE offers investment opportunities without ignoring shariah compliance; this also increases the CE of IB (Nawaz and Haniffa 2017).

This study uses the VAIC method for IC measurement. The advantage of this is that the VAIC value and its dimensions can be calculated using data obtained from the company's financial statements, which are generally easily found in published financial reports. VAIC provides standard and consistent measures, so it is more objective than other methods. The use of financial data facilitates the calculation of each indicator ratio for comparison with other indicators (Hang Chan 2009). Furthermore, obtaining and calculating data is uncomplicated because the calculations use standard formulas and, thus, allow comparison between companies (Maditinos et al. 2011). Nevertheless, the VAIC method has also been criticised. Stahle et al. (2011) argue that the VAIC approach has nothing to do with IC since it only signifies the labour and efficiency of the company's capital investment. Another criticism of this approach is that it conflicts with the accounting concept in the case of calculating HC and SC (Andriessen 2004). Iazzolino and Laise (2013) respond to this rebuttal and argue that the VAIC approach does not conflict with accounting principles. This study adopted VAIC to determine the investment efficiency of HC, SC,



and CE. VAIC, as an IC measuring instrument, is easily applicable to Islamic banks with various limitations on the company's internal information disclosure.

The VAIC value of a company is its total HCE, SCE, and CEE. The value added (VA) is here specifically defined as the added value for IBs. In other words, it results from reducing the input–output ratio for the operation of the IB. This amount is taken from the bank's operating profit plus salary costs and represents added value because operating profit reduces all costs from income (Tran et al. 2020). The current literature links employee productivity with the intellectual level so that labour cost indicators can be used to calculate the VA. The intellectual development of employees increases labour productivity. In other words, the VA in labour costs reflects the impact of human resources in creating added value (Iazzolino and Laise 2013). The formula for calculating the VA is profit before tax plus salary costs. HCE is calculated by dividing VA by HC, which is proxied by personal costs (Tran et al. 2020).

### 2.2. Previous Research on Intellectual Capital and Performance in the Banking Industry

Nawaz (2017) and Ousama et al. (2019) empirically prove that high IC efficiency boosts the profitability of IBs. This study uses data of 47 IBs assesses profitability using their ROA. Nawaz and Haniffa (2017) relate IC and the profitability of Islamic financial institutions (IFIs) for a sample of 68 IFIs in 18 countries. In part, the results show that HCE and CEE have a significant positive effect on ROA. Nawaz (2019) examines 47 IBs from various countries in the period 2005–2010 and finds that HC had a significant positive impact on their market value both before the crisis and after the crisis. Ousama et al. (2019) report that IC has a positive effect on the performance of IBs, although this effect is smaller to that in other studies. Furthermore, HCE and CEE have a positive effect on IB performance, while SCE has an insignificant impact. This research was conducted on IB during 2011–2013 in the GCC countries.

Several studies analyse the relationship between IC and the performance of commercial banks. Duho (2020) examines banking in Ghana in 2012–2017 and shows that HC is the main factor driving banking efficiency through the slack-based technical efficiency method. Soewarno and Tjahjadi (2020) report that IC measured by VAIC and the adjusted value-added intellectual coefficient (A-VAIC) is related to the performance of the banking sector in Indonesia with the proxies of ROA, ROE, asset turnover (ATO), and price-to-book value (PBV). Mahmood et al. (2018) analyse the relationship between IC and market performance in the banking sector in Pakistan using a questionnaire.

Islamic banks are growing rapidly in various countries and are part of the knowledge-based industry. Therefore, IBs will achieve better performance if they continue to manage ICs more efficiently. IB performance has been explored by many previous researchers based on market-based and accounting measurements. This paper attempts to fill the gap by exploring the IC efficiency of IBs and their relationship to performance on the basis of IFIB. The findings of this study will be useful for Islamic banks who want to measure IC efficiency; thus, it will motivate managers to measure and manage their IC.

Several studies discuss the link between IC and performance using accounting (ROA and ROE) and market-based (Tobin's Q) measures. IBs exist as for-profit financial institutions in addition to carrying out social functions. Therefore, the measurement of the performance of IBs should include financial and social factors. Aziz and Mohamad (2016) argue that Islamic social business in IBs can use measures of income from mudarabah and murabaha contracts as well as the distribution/payment of zakat from IBs. Zakat is a mandatory contribution based on income or asset ownership that meets the requirements and is distributed to certain groups that have been written in the Qur'an, for example, the poor, travellers, in debt, slaves, and others (Abdullah et al. 2013). Previous studies do not use measures that capture the uniqueness of these IBs. Therefore, this research combines financial and social performance measurements to address the links between performance and IC.

## 3. Hypothesis Development and Methodology

### 3.1. Conceptual Framework and Hypothesis Development

The resource-based view (RBV) of strategic management was developed by companies to provide options in practical decision making. Numerous strategic management studies deal with how companies can achieve a competitive advantage. Therefore, a consistent method of analysis is needed to identify the characteristics of RBV comprehensively and objectively (Nagano 2019). RBV treats the company as a collection of unique resources (such as IC) that are useful for developing products, services, and strategies (Barney 1991). Nonaka et al. (2017) argue that companies need to create new knowledge to ensure their future existence, so they must have good strategic and resource management. Several previous studies analyse the IC components independently of their dependent variables, such as research (Akkas and Asutay 2022; Chatterjee et al. 2022; Dalwai and Mohammadi 2020; Dalwai and Salehi 2021; Kweh et al. 2021; Vo and Tran 2021). This opinion considers each component's specific impact on performance rather than the impact of IC more broadly. This study agrees with previous researchers that the literature treats the IC components, namely, HC, SC, and CE, as independent variables.

### 3.1.1. IC and IB Performance

The added value of IC can be an indicator of IB performance because, together, HC, SC, and CE can generate a competitive advantage, which in turn improves performance. Therefore, IBs must increase their IC investment because it has the potential to improve the sustainability of their performance in the long run. According to Cenciarelli et al. (2018), IC performance is useful for achieving above-average performance, which in the long run maintains company stability. The results of Nawaz and Haniffa (2017), Ozkan et al. (2017), and Tiwari and Vidyarthi (2018) support this argument that IC has a positive relationship with ROA. By contrast, Tran and Vo (2018) find that, statistically, IC is not significantly associated with ROA. Based on these arguments that IC can also improve the performance of IBs, the first hypothesis of this study is:

**Hypothesis 1 (H1).** *IC has a positive effect on the performance of IBs.*

### 3.1.2. Human Capital and IB Performance

IBs must maximise their use of HC for the maximum probability of success. There is much empirical evidence that supports the view that HC positively affects company performance. Ting and Lean (2009) find evidence that the efficiency of HC has a significant positive effect on the profitability of financial institutions. Tiwari and Vidyarthi (2018) and Nawaz and Haniffa (2017) prove that HC has a positive and significant impact on the performance of IFIs and commercial banks in India. Meanwhile, McDowell et al. (2018) find a positive relationship between HC and organisational performance. Bontis et al. (2018) note that HC contributes to explaining economic performance, which is positively influenced by the presence of postgraduate employees and increased added value per employee.

Several other studies contradict these results. Chu et al. (2011) find that HC has a negative relationship with market performance. The results of Maditinos et al. (2011) show that HC has no significant effect on ROA and company growth. This study was conducted using a sample of 96 companies listed on the Athens Stock Exchange for the period 2006–2008. Firer and Williams (2003) find a significant negative relationship between HC and firm productivity as proxied by asset turnover and stock market performance.

On the basis of the literature review above, we can summarise that HC consists of the attributes of knowledge, expertise, skills, experience, competence, creativity, loyalty, problem-solving skills, and motivation. All of these attributes are inherent in the human being concerned and must be maintained and developed to be able to generate new ideas and innovation. In the context of IBs, HC is the most crucial aspect of the organisation in developing and creating new banking products and appealing to customers through

adherence to sharia principles; those who are employed by IBs are required to have a broader range of capabilities and expertise than in conventional banks, considering that IB products are more complex.

IBs need to develop and improve the capabilities of their employees. They need to focus on the development of soft skills, which may include training that increases their ability to develop new marketing, product, and customer strategies. In addition, they need to maintain existing employee skills and knowledge, for example, by continuously upgrading their product knowledge in light of the latest developments. Special education about Islamic character is also critical, considering that the core business of IBs is closely related to the teachings of Islam. Relevant resources must be developed and utilised optimally so that, in the end, IBs reach their performance goals both in financial terms and in sharia observance. The literature shows that HC has a positive relationship with performance, and the second hypothesis is, thus:

**Hypothesis 2 (H2).** *HC has a positive effect on the performance of IBs.*

### 3.1.3. Structural Capital and IB Performance

Nawaz and Haniffa (2017) find a positive but insignificant relationship between the efficiency of SC and ROA in a study of 64 IFIs in 18 countries (across Asia, Europe, and the Middle East) from 2007 to 2011. Tiwari and Vidyarthi (2018) produce different results and find that SC is positively related to bank performance in a study of 39 banks listed on India's stock exchange from 1999 to 2015. McDowell et al. (2018) survey 460 SMEs in the United States from 2015 to 2016. They find that SC and organisational performance have a positive relationship. Novas et al. (2017) state that SC has a positive and significant relationship with organisational performance. Hamdan (2018) proves that SC has a positive and significant impact on company performance in Saudi Arabia and Bahrain. Bontis et al. (2000) identify a positive and significant relationship between SC and business performance.

Some research results contradict previous research that SC hurts company performance. For example, Palazzi et al. (2019) finds that SC hurts company performance due to the inefficient use of resources. This is consistent with the finding of Bontis et al. (2018) that SC does not affect company performance.

In the context of IBs, SC is the infrastructure that supports HC in IBs and allows assets to be distributed through a company's structures and processes. SC comes from the knowledge within HC in relation to infrastructure, data, standard operating practices, strategies, and corporate culture that support the business processes of IBs and the creation of patents and trademarks that support HC functions. If all SC is managed and utilised optimally, it can lead to the creation of new markets or the maintenance of existing markets. Customers are attracted by the ease of service and product choices, which in turn create customer satisfaction and loyalty, and IBs' performance objectives can then be easily achieved. Based on this, the third hypothesis is as follows:

**Hypothesis 3 (H3).** *SC has a positive effect on the performance of IBs.*

### 3.1.4. Capital Employed and Islamic Bank Performance

CE supports IC in creating added value through improved customer service. Therefore, IBs need to increase the efficiency of CE to improve their performance. This is supported by the finding of Kweh et al. (2019), Nawaz and Haniffa (2017), and Tran and Vo (2018) that show that CEE has a positive relationship with performance.

CE reflects the company's physical and financial ability to create added value by increasing the efficiency of its assets. IBs must empower all physical and financial assets to increase profitability, manage risk, and thereby reduce the cost of financing. Therefore, more efficient use of CE will increase the performance of IBs, and the fourth hypothesis is as follows:

**Hypothesis 4 (H4).** *CE has a positive effect on the performance of IBs.*

*3.2. Methodology*

3.2.1. Data Sources

This study uses a sample of 94 IBs drawn (after checking for outlier) from a global sample of 96 and data obtained from the BankScope database for the period 2009–2019. This research data type is a quarterly time series with an unbalanced panel model. The sample is selected based on the criteria of non-window banking Islamic banks.

3.2.2. Dependent Variables

The profitability of IBs is the dependent variable in this study. ROA is used as an accounting-based measure of IB performance. This measure follows previous studies (Al-Musali and Ismail 2016; Nawaz and Haniffa 2017; Ozkan et al. 2017; Tiwari and Vidyarthi 2018). The ROA is net income divided by total assets. The performance measure IFIB references income attributable to IB products (Grassa et al. 2018), such as financing instruments and profit-sharing distribution (Mrad and Mateev 2020). The formula for measuring this variable is the natural log of total income from murabaha, musyarakah, mudharabah, ijarah, istisna, salam, and wakalah financing instruments. Murabaha is a sale and purchase contract that charges a margin agreed by the Islamic bank and the customer (Choudhury and Alam 2013). Musyarakah is a joint venture between an Islamic bank and its customers. Mudharabah is a contract between an Islamic bank as an investor and a project manager (Abdullah et al. 2013). Ijarah is a leasing contract; meanwhile, istishna is a sale and purchase contract in which the goods are produced and shipped in the future (Al Rahahleh et al. 2019). Salam is a purchase with deferred delivery referring to the contract price, commodity, quantity, quality, and delivery date. Furthermore, wakalah is an agency that acts on behalf of the contract subject, for example, services for letters of credit, bill collection, and fund management (Hasan and Dridi 2011).

3.2.3. Independent Variables

IC variables are measured using the VAIC method, adopted from Pulic (2004), which defines VAIC as comprising HCE, SCE, and CEE. HCE is an employee cost that is treated as an investment rather than an expense. HCE shows the amount of VA that can be generated in monetary units from investment in HC (Tran and Vo 2018). The first step of the VAIC method is calculating the VA from the amount of profit before tax plus personal cost (HC). After obtaining the VA value, the company's HCE is calculated as follows:

$$HCE = VA/HC$$

SCE includes factors that positively affect employee productivity, such as software and hardware, trademarks, patents, and other assets (Dženopoljac et al. 2016). SC is obtained from VA minus HC. SC considers all value created apart from the contribution of human resources such that the SCE represents the efficiency value of the SC. SCE is calculated as follows:

$$SCE = SC/VA$$

CEE is the efficiency of the physical and financial capital used by the company. CEE is calculated by dividing VA by net assets as follows:

$$CEE = VA/CE$$

3.2.4. Control Variable

The diversity of IBs in the sample has an impact on the findings, so a control variable is needed. Proxies for control variables commonly used in studies on the same topic are firm size and leverage (Nawaz and Haniffa 2017; Ozkan et al. 2017; Tiwari and Vidyarthi 2018). This study uses firm size as a control variable (the log of total assets), leverage, which is

proxied by the ratio of total debt to total assets, and gross domestic product (GDP) growth, which measures the growth of GDP and is growth in year t minus year t−1 (Shawtari 2018).

3.2.5. Regression Model

This regression model tests the statistical relationship between IC and IB performance. To test the research hypotheses established, we estimate four regression models. Models 1 and 2 examine the relationship of IC in the aggregate to the performance of IBs using ROA and IFIB as proxies for performance. Models 3 and 4 examine the relationship between each of the components of IC, namely, HCE, SCE, and CEE, and the performance of IBs with ROA and IFIB as proxies. Variables to control size and leverage increase the explanatory power in each model (Tiwari and Vidyarthi 2018). The regression model for this study is as follows:

Model 1:

$$ROA_{it} = \beta_0 + \beta_1 VAIC_{it} + \beta_2 SIZE_{it} + \beta_3 LEV_i + \varepsilon_{it}$$

Model 2:

$$IFIB_{it} = \beta_0 + \beta_1 VAIC_{it} + \beta_2 SIZE_{it} + \beta_3 LEV_i + \varepsilon_{it}$$

Model 3:

$$ROA_{it} = \beta_0 + \beta_1 HCE_{it} + \beta_2 SCE_{it} + \beta_3 CEE_i + \beta_4 SIZE_{it} + \beta_5 LEV_{it} + \varepsilon_{it}$$

Model 4:

$$IFIB_{it} = \beta_0 + \beta_1 HCE_{it} + \beta_2 SCE_{it} + \beta_3 CEE_i + \beta_4 SIZE_{it} + \beta_5 LEV_{it} + \varepsilon_{it}$$

## 4. Empirical Results

A descriptive analysis for each variable studied is presented in Table 1. All data consist of 94 IBs from 24 countries in the sample with an observation period of 11 years, and a quarterly data period with an unbalanced panel model. VAIC is the accumulated HCE, SCE, and CEE. The average VAIC score was 3.77177; the most significant component supporting VAIC was HCE. The average HCE score reached 3.02925, which is much higher than the average SCE and CEE scores of 0.64156 and 0.10095, respectively. The highest IC score of 29.65699 was obtained for the IB of AmBank Islamic Berhad in Malaysia. In comparison, the lowest score of −9.20528 was obtained for the IB of the First National Bank Modaraba in Pakistan. These results are consistent with research by IFIs in various countries (Nawaz and Haniffa 2017). The observations pertaining to performance show that overall, the performance of IBs is positive in terms of ROA and IFIB with an average score of 3.51144 and 3.55915. This means that during the observation period, IBs generated profits and income from Islamic contracts.

**Table 1.** Descriptive statistics.

| Variable | Obs | Mean | Std. Dev. | Min | Max |
|---|---|---|---|---|---|
| ROA | 1645 | 3.51144 | 5.35694 | −9.50281 | 24.84503 |
| IFIB | 1463 | 3.55915 | 1.73684 | −0.68888 | 7.24537 |
| VAIC | 1645 | 3.77177 | 4.38699 | −9.20528 | 29.65699 |
| HCE | 1645 | 3.02925 | 4.14367 | −9.71000 | 28.60676 |
| SCE | 1645 | 0.64156 | 1.31523 | −6.33333 | 15.55696 |
| CEE | 1645 | 0.10095 | 0.14744 | −3.98160 | 0.60244 |
| SIZE | 1645 | 7.49031 | 2.03844 | 0.77198 | 11.02267 |
| LEV | 1645 | 0.90498 | 0.27010 | −2.18797 | 1.19673 |
| GDP | 1645 | 4.14189 | 2.33832 | −2.76770 | 11.95656 |

The average coefficient of the control variable size is 7.49031, with a standard deviation of 2.03844. Size is measured by total assets, and the standard deviation shows that there is not a wide range in IB size. The second control variable, leverage, measures the level

of risk faced by an IB. Overall, the average leverage is 0.09498, with a minimum value of $-2.18797$ and a maximum value of 1.19673. This shows the diversity of risks in various IB. The third control variable, average GDP growth, is 4.14189, with a minimum value of $-2.76770$ and a maximum value of 11.95656.

Table 2 sets out the results of the panel data regression with fixed effects. All sample data were entered into the test. In total, there are eight models with two dependent variables, namely, ROA and IFIB. Models 1 and 2 connect VAIC to ROA and IFIB performance. VAIC has a positive and significant relationship with ROA ($p < 0.01$) and IFIB ($p < 0.1$); H1 is supported. Models 3 and 4 link HCE with ROA ($p < 0.01$) and IFIB ($p < 0.1$), both of which show a positive and significant relationship; H2 is supported. Models 3 and 4 relate SCE to ROA and IFIB. The result shows that SCE has a significant positive effect on ROA ($p < 0.05$), while SCE with IFIB has a negative but not significant effect; this statistical result supported H3. H4 was tested with Models 3 and 4, and the results show that CEE has a significant positive relationship to ROA and IFIB ($p < 0.01$); H4 is supported.

**Table 2.** Panel A: full sample.

| Variable | Model 1 ROA | Model 2 IFIB | Model 3 ROA | Model 4 IFIB |
|---|---|---|---|---|
| VAIC | 0.170 *** | 0.003 * | | |
| | (0.027) | (0.016) | | |
| SIZE | 1.942 *** | 0.883 *** | 1.797 *** | 0.861 *** |
| | (0.134) | (0.053) | (0.132) | (0.053) |
| LEV | $-2.529$ *** | 0.252 | $-0.521$ *** | 0.257 |
| | (0.628) | (0.412) | (0.609) | (0.409) |
| GDP | $-0.065$ | 0.027 | $-0.068$ * | 0.024 |
| | (0.040) | (0.023) | (0.039) | (0.023) |
| HCE | | | 0.124 *** | 0.017 * |
| | | | (0.030) | (0.019) |
| SCE | | | 0.101 ** | 0.012 |
| | | | (0.060) | (0.029) |
| CEE | | | 6.333 *** | 0.503 *** |
| | | | (0.591) | (0.163) |
| _cons | $-8.824$ *** | $-3.202$ *** | $-8.246$ *** | 3.074 *** |
| | (0.977) | (0.489) | (0.955) | (0.489) |
| N | 1645 | 1463 | 1645 | 1463 |
| F Statistic | 48.45 | 319.52 | 52.51 | 20.99 |
| *p*-value F Statistic | 0.0000 | 0.0000 | 0.0000 | 0.0000 |
| R Square | 0.387 | 0.687 | 0.403 | 0.690 |

Standard errors in brackets. * $p < 0.1$, ** $p < 0.05$, *** $p < 0.01$.

Overall, H1, H2, H3, and H4 are accepted, SCE has no significant impact on IFIB performance. VAIC shows added value for IBs, thereby increasing performance. HCE, SCE, and CEE also have some positive effects on the performance of IBs. These results are consistent with previous studies (Ousama et al. 2019; Cenciarelli et al. 2018; Ozkan et al. 2017; Nawaz and Haniffa 2017; Tran and Vo 2018). These empirical results support the RBV theory that in the context of IB, the composition of assets (both tangible and intangible) can boost company performance. IB managers must consider the composition of assets in every investment decision. HC assets are most crucial because it is challenging to duplicate human capabilities, and employees may be recruited away by other companies by the inducement of higher compensation.

The results of this study provide empirical evidence that the IC can significantly support IB performance, and IC components separated into HC, SC, and EC provide the same result. Therefore, banks must pay attention to the competence of their employees not only at the time of recruitment, but throughout their employment; HC skills require maintenance and upgrading to provide optimal added value. IBs must budget for personal

expenses related to upgrading and honing skills through training or other activities. Future technological developments will change the work usually performed by humans. Some jobs will be replaced by technology, including robots, leaving humans to master specific jobs that cannot be replaced by technology.

Policymakers, such as the Sharia Supervisory Board, must know about developments in other industries to avoid overlapping fatwas and rigidity in providing Islamic legal advice regarding operations. IBs offer complex products and schemes, and this sector is associated with a range of industries seeking funding. It is, thus, crucial to understand the core business and risk profile and stay within sharia principles. In addition to having good selling skills, those working in sales roles for IBs must be equipped with adequate religious understanding. The main sales task is marketing the IBs' products, but there is an underlying message that must be delivered to customers and investors. Sales staff must be able to communicate to the public that transactions in IBs are not just part of business, but encompass religious values that must be honoured.

IBs must also pay attention to SC as an infrastructure that supports HC and CE. In general, IBs have a low level of SC compared to standard commercial banks, especially in developing countries. SC that comes from the adoption of technology, the deposition of internal processes such as corporate culture, and internal procedures allow IBs to reduce both internal and external shocks to the company.

Robustness tests are carried out by separating the results by region; this is performed because each region has a different level of assets and different cultures, including in respect of risk management. We thus divided the sample into regional sub-samples, namely, the Middle East and North Africa (MENA) and Non-MENA. The statistical results are presented in Table 3 for the MENA region and Table 4 for Non-MENA.

**Table 3.** Panel B: MENA.

| Variable | Model 1 ROA | Model 2 IFIB | Model 3 ROA | Model 4 IFIB |
|---|---|---|---|---|
| VAIC | 0.425 *** | −0.006 | | |
| | (0.063) | (0.032) | | |
| SIZE | 2.832 *** | 0.977 *** | 2.535 *** | 1.012 *** |
| | (0.192) | (0.077) | (0.180) | (0.070) |
| LEV | −4.331 *** | 3.108 ** | −4.793 *** | 0.794 |
| | (1.043) | (1.257) | (0.900) | (1.148) |
| GDP | −0.032 | 0.019 | −0.083 * | −0.003 |
| | (0.054) | (0.028) | (0.045) | (0.024) |
| HCE | | | −0.192 ** | −0.135 *** |
| | | | (0.075) | (0.039) |
| SCE | | | 0.205 ** | −0.035 |
| | | | (0.084) | (0.043) |
| CEE | | | 38.806 *** | 6.507 *** |
| | | | (1.872) | (0.627) |
| _cons | 14.158 *** | −6.867 *** | −13.330 *** | −5.359 *** |
| | (1.360) | (1.108) | (1.289) | (0.999) |
| N | 906 | 805 | 906 | 805 |
| F Statistic | 44.64 | 22.17 | 134.58 | 46.96 |
| *p*-value F Statistic | 0.0000 | 0.0000 | 0.0000 | 0.0000 |
| R Square | 0.523 | 0.674 | 0.599 | 0.723 |

Standard errors in brackets. * $p < 0.1$, ** $p < 0.05$, *** $p < 0.01$.

**Table 4.** Panel C: non-MENA.

| Variable | Model 1<br>ROA | Model 2<br>IFIB | Model 3<br>ROA | Model 4<br>IFIB |
|---|---|---|---|---|
| VAIC | 0.107 *** | −0.013 | | |
| | (0.019) | (0.017) | | |
| SIZE | 0.713 *** | 0.874 *** | 0.572 *** | 0.860 *** |
| | (1.130) | (0.083) | (0.125) | (0.087) |
| LEV | −0.653 | −0.094 | −0.602 | −0.074 |
| | (0.510) | (0.446) | (0.493) | (0.449) |
| GDP | 0.103 * | 0.023 | 0.122 ** | 0.024 |
| | (0.053) | (0.041) | (0.051) | (0.041) |
| HCE | | | 0.107 *** | −0.020 |
| | | | (0.106) | (0.020) |
| SCE | | | −0.009 | −0.015 |
| | | | (0.051) | (0.035) |
| CEE | | | 2.584 *** | 0.197 |
| | | | (0.364) | (0.162) |
| _cons | −3.339 *** | −2574 *** | −2.723 *** | −2.517 *** |
| | (0.910) | (0.660) | (0.871) | (0.681) |
| N | 739 | 658 | 739 | 658 |
| F Statistic | 10.91 | 11.87 | 17.05 | 8.02 |
| *p*-value F Statistic | 0.0000 | 0.0000 | 0.0000 | 0.0000 |
| R Square | 0.345 | 0.705 | 0.318 | 0.707 |

Standard errors in brackets. * $p < 0.1$, ** $p < 0.05$, *** $p < 0.01$.

Table 3 shows that VAIC has a positive effect on the performance of ROA ($p < 0.01$), but it is not significant for IFIB ($p > 0.1$). HCE, SCE, and CEE have a significant positive effect on ROA, while HCE has a significant positive effect, though SCE does not, for IFIB proxies.

Table 4 shows that IC has a positive effect on performance only on the proxy for ROA ($p < 0.05$). CEE only has a significant positive effect when associated with ROA ($p < 0.01$), while SCE is significantly negative for the proxy of ROA performance. HCE has no significant effect on any of the performance proxies.

## 5. Conclusions, Limitations, and Recommendations

This paper analysed the IC efficiency of Islamic banks with performance on the basis of ROA and IFIB using data from 94 IBs from 24 countries. This study aimed to understand the effects of IC, both total and partial, on IB performance, which is proxied by ROA and IFIB. The empirical results show some important findings. First, VAIC has a significant positive effect on IB performance using both ROA and IFIB proxies. Second, HCE has a significant positive effect on ROA and IFIB. In addition, CEE has a significant positive relationship with ROA and IFIB. Finally, the study also revealed that SCE has a significant positive on ROA, but no relationship with IFIB.

The association between IC and tangible assets can create added value for IBs and generate a competitive advantage. IBs need a competitive advantage to survive in the global banking industry. The main competitors of IBs are commercial banks and financial technology; competition is very fierce in relation to ease of access and technology-based services. IBs have specific features and allowed forms of agreement and requirements for sharia compliance, such as honesty, not investing in illicit industries, and prohibiting speculation. IBs must be able to convert this particularity into a competitive advantage. IB can make this happen through HC development. CE is a tool used by humans in achieving the best performance. IBs with high tangible assets and human professionalism will generate maximum profit. CE fulfils regulatory requirements and capital for market penetration

The above findings can provide several implications. First, management is clearer in understanding the effect of IC on IB performance. Second, the analysis of the HCE, SCE, and CEE coefficients provides managers with guidance to identify the main drivers of the

VAIC coefficients. Third, the findings of this study have important implications for IBs, because IC is a driver of IB performance that contributes to economic growth. Furthermore, these findings are consistent with previous studies in various industrial areas. Therefore, IBs should encourage IC investment to improve economic performance and growth. Thus, this research can be used as a reference by policy makers in formulating IC development policies in the IB sector in the future. Finally, IC is a company's strategic resource that adds value. Therefore, it needs to be disclosed in the annual report to give a positive signal to investors. Therefore, policy-making authorities should take appropriate steps to ensure that IBs disclose additional information regarding their ICs and provide full disclosure to investors.

This study has several limitations. First, the sample used is limited to IBs, and further research could be pursued using IFIs. Second, this study uses the VAIC method to measure IC's contribution to value creation, and other methods could be used to approximate IC, for example, content analysis. Third, sharia-based performance proxy uses IFIB, and this measure is limited in scope to sharia contract financing income and cannot cover compliance with other sharia principles, for example, safeguarding the value of human life, society, and wealth. Further research can use maqasid shariah as a more comprehensive measurement to improve the weaknesses of IFIB.

**Author Contributions:** Conceptualization, P.P., W.Y., T.F. and M.S.; methodology P.P., W.Y., T.F. and M.S.; software P.P. and T.F.; validation, W.Y. and M.S.; formal analysis P.P., W.Y. and T.F.; investigation, W.Y. and M.S.; resources, P.P.; data curation, P.P. and T.F.; writing—original draft preparation, P.P. and T.F.; writing—review and editing W.Y. and M.S.; visualization, P.P.; supervision, W.Y., T.F. and M.S. project administration, P.P. All authors have read and agreed to the published version of the manuscript.

**Funding:** We gratefully acknowledge the funding support for APC provided by Padjadjaran University, Bandung 40132, Indonesia.

**Institutional Review Board Statement:** Not applicable.

**Informed Consent Statement:** Not applicable.

**Data Availability Statement:** Not applicable.

**Conflicts of Interest:** The authors declare no conflict of interest.

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
