# Peer review of "Cross-Region Comparison Intellectual Capital and Its Impact on Islamic Banks Performance"

_economies, doi:10.3390/economies10030061_

Round 1

Reviewer 1 Report

Dear authors,

The analysis is correct and reflects an interesting subject of the search for factors having the strongest impact on a Bank's efficiency.

The critics of the VAIC approach and methodology are reflected in the paper, but the influence of these critics on the research is not discussed.

The article presents a very interesting analysis and results about checked correlations between the components of intellectual capital and the outcomes and performance of the Banks.

Nevertheless, the clear lack of understanding of the following components: how the methodology (based on accounting data) has limited the results obtained by researchers; how the sample was chosen; what criteria permitted to entitle the investigated banks as Islamic banks (only the national citizenship and territory of registry of a bank? the religion in the area of national countries does it determine any common features of the banks? what kinds of characteristics are common for the sample?).

The absence of these answers could be compensated with the information in Appendices, but there are no annexed documents, tables, maps, diagrams nor any description of criteria of sampling.

There are some doubts concerning the content and the form.

Substantial concerns include:

1 -  terms use: value-added IC-exchange is not abbreviated as VAIC (Value Added Intellectual Coefficient), which is mentioned in the first lines of the Abstract. (Sentence: "...  a value-added IC-exchange method known as the
VAIC.") Is it a systemic confusion or just a typo?

2 - what are the specific features of Islamic banks?  The third paragraph describes the Muslim religion's requirements applied to banking activities. But for the purpose of this article, how the banks were identified as Islamic or not?

3 - How the 39 banks included in the research were chosen among 96 mentioned as a global sample (3.2.1.)?

4 - the components of the intellectual capital (IC) are discussed, the relational capital is mentioned twice, reputation and customers, as well and innovation - are mentioned, but the authors do not explain why these components are not examined in their research. Yes, these components are not presented in the financial reporting, in accounting, but the readers should guess themselves.

5 - in February 2022, the analysis of data of 2009-2018 years raises questions about the last 3 years (2019-2021) - in 2020-2021 the results could be influenced by the pandemic more than by other variables (this influence would represent a specific interest for the study), but what about 2019, why 2019 is omitted? 

Formal issues:

1 - the terms of mudarabah, zakat (2.2.), murabaha, musyarakah, mudharabah, ijarah, istisna, salam and wakalah (3.2.2.) - should be defined and described. 

2 - does the abbreviation "GCC" relate to the Gulf Cooperation Council? It should be decoded.

3 - in the sentence the hypotheses' numbers are mixed: "Overall, H1, H3 and H4 are accepted, and H2, connecting SCE with IB performance, is rejected" - the previous text about the results and their discussions shows that hypotheses 1, 2 and 4 are supported, the 3rd one is rejected (H3 relates SC to performance, the analysis shows negative or non-significant effect).

These remarks are the first ones, only several comments from the first version of the article.

I would be glad to read the corrected text of the article at the next step of editing and reviewing.

Reviewer 2 Report

The paper investigates the issue of IC’s effects on the performance of IBs and contributes to literature especially through the cross-country long-term approach and assessment of social performance reflecting the compliance with the sharia rules. As it stands, the paper could be suitable for publishing provided that some revisions are performed:  

  • While ROA and ROE are widely known measures of financial performance, the authors could consider presenting in more detail their measure of social performance related to sharia aspects. Also, terms like “mudarabak”, „murabaha”, „zakat” etc. might not be well understood by the international reader and should be further explained in a footnote. This might be useful especially since this issue is not widely debated and approached in the literature.
  • I don’t find section 2.3 to be justified as a distinct section, since it does not bring a significant amount of new information on a distinct aspect/issue. It could be no more than a conclusion to section 2.
  • When describing the data, it might be useful to know from how many countries are the 39 IBs coming and what is the distribution of the sample per country. This might also be helpful for assessing appropriateness of the estimation method. Also, how were the 39 banks selected from the total of 96?
  • The number of controls is quite small. Have the authors tried to test the robustness of their results against the introduction of other control variables? (macro-economic variables, widely available, might be a good way to start).
  • The authors might want to check for correlations of VAIC with HCE, SCE and CEE. When the last three variables are introduced in models 3 and 4 (Table 2), VAIC losses its statistical significance. Maybe it would be a good idea to consider introducing them separately in the regression model.
  • It is common practice for acronyms to be explained the first time they appear in text, and some are not (like GCC countries, ATO, PBV, etc.).

Reviewer 3 Report

Recommendations for the authors of the article:

  1. In the "abstract", the structure of the article should be described in a few sentences.
  2. The concept of R. Hall (1992, 1993), A. Brooking (1996, 1997), B. Lowendahl (1997) and M. Litsch, A. Markom, S. Schunder (2006) should be added to the "literature review" section. The views of researchers can be particularly associated with the concept of human capital used in financial institutions.
  3. In the methodology, please provide research limitations.
  4. Please correct especially the last section of the article (5). I propose to present the conclusions from the research in the form of points.

Reviewer 4 Report

I have carefully read your manuscript "CROSS-REGION COMPARISON INTELLECTUAL CAPITAL AND ITS IMPACT ON ISLAMIC BANKS PERFORMANCE" and I enjoyed it. 

I think that the manuscript is well written and structured, addressing an interesting and important topic.

However, I believe that to accept the paper, the authors still need to focus on some changes or justify some choices underlying the research:

the authors should explain the adoption of value-added intellectual coefficient (VAIC) and why they did not use other methods

authors should better specify the composition of the sample

why was the time period 2009-2018 chosen?

the contribution in terms of originality and innovativeness of the authors should be more explicit.

scientific implications should be more explicit in the last paragraph

institutional and managerial implications should be more explicit in the introduction

out of 60 references, there are only 2 from 2020 onwards... it seems to me to be very poorly adjusted

All the best! And good luck with your research

Round 2

Reviewer 1 Report

Hi, 

thank you for your corrections and your letter!

It is much better now, and I think it is time to publish )

Good luck?

Reviewer 2 Report

The authors have done a good job in revising the paper and I recommend it to be published in "Economies" in its present form.

Reviewer 3 Report

Thrush Authors

I believe that this version of the article is scientifically and methodologically answered.

I wish you successes